# Salicylic Acid as a Salt Stress Mitigator on Chlorophyll Fluorescence, Photosynthetic Pigments, and Growth of Precocious-Dwarf Cashew in the Post-Grafting Phase

**DOI:** 10.3390/plants12152783

**Published:** 2023-07-27

**Authors:** Thiago Filipe de Lima Arruda, Geovani Soares de Lima, André Alisson Rodrigues da Silva, Carlos Alberto Vieira de Azevedo, Allesson Ramos de Souza, Lauriane Almeida dos Anjos Soares, Hans Raj Gheyi, Vera Lúcia Antunes de Lima, Pedro Dantas Fernandes, Francisco de Assis da Silva, Mirandy dos Santos Dias, Lucia Helena Garófalo Chaves, Luciano Marcelo Fallé Saboya

**Affiliations:** 1Post Graduate Program in Agricultural Engineering, Federal University of Campina Grande, Campina Grande 58430-380, Brazil; thiago.filipe@estudante.ufcg.edu.br (T.F.d.L.A.); andre.alisson@estudante.ufcg.edu.br (A.A.R.d.S.); carlos.alberto@professor.ufcg.edu.br (C.A.V.d.A.); allesson.ramos@estudante.ufcg.edu.br (A.R.d.S.); hans.gheyi@ufcg.edu.br (H.R.G.); vera.lucia@professor.ufcg.edu.br (V.L.A.d.L.); pedro.dantas@professor.ufcg.edu.br (P.D.F.); francisco.assis@estudante.ufcg.edu.br (F.d.A.d.S.); mirandy.santos@estudante.ufcg.edu.br (M.d.S.D.); lucia.garofalo@professor.ufcg.edu.br (L.H.G.C.); luciano.saboya@professor.ufcg.edu.br (L.M.F.S.); 2Academic Unit of Agrarian Sciences, Federal University of Campina Grande, Pombal 58840-000, Brazil; lauriane.soares@professor.ufcg.edu.br

**Keywords:** *Anacardium occidentale* L., plant hormone, salt stress

## Abstract

Salicylic acid is a phytohormone that has been used to mitigate the effects of saline stress on plants. In this context, the objective was to evaluate the effect of salicylic acid as a salt stress attenuator on the physiology and growth of precocious-dwarf cashew plants in the post-grafting phase. The study was carried out in a plant nursery using a randomized block design in a 5 × 4 factorial arrangement corresponding to five electrical conductivity levels of irrigation water (0.4, 1.2, 2.0, 2.8, and 3.6 dS m^−1^) and four salicylic acid concentrations (0, 1.0, 2.0, and 3.0 mM), with three replications. Irrigation water with electrical conductivity levels above 0.4 dS m^−1^ negatively affected the relative water content in the leaf blade, photosynthetic pigments, the fluorescence of chlorophyll *a*, and plant growth and increased electrolyte leakage in the leaf blade of precocious-dwarf cashew plants in the absence of salicylic acid. It was verified through the regression analysis that salicylic acid at a concentration of 1.1 mM attenuated the effects of salt stress on the relative water content and electrolyte leakage in the leaf blade, while the concentration of 1.7 mM increased the synthesis of photosynthetic pigments in precocious-dwarf cashew plants.

## 1. Introduction

Belonging to the family *Anacardiceae*, cashew (*Anacardium occidentale* L.) is a widely cultivated species in the semi-arid region of northeastern Brazil due to its high commercial and socioeconomic value, serving as an important source of employment and income for the population [1]. Cashews are rich in vitamin C, calcium, phosphorus, and iron, and play a key role in the fruit-processing industry, giving origin to products such as sweets and juices, cashew nuts, and cashew bagasse, which is used to produce flours, animal food, and medications [2,3].

Brazil is one of the principal cashew nut producers in the world, with a production of 111,103 tons in 427,144 hectares. In 2021, the Northeast region of Brazil produced approximately 106,675 tons per hectare of cashew nuts, with the State of Ceará ranking as the largest cashew-producing state with 62,977 tons. Piauí and Rio Grande do Norte stand out as second and third place, producing 19,020 and 16,920 tons per hectare, respectively [4]. In the same year, the State of Paraíba produced approximately 670 tons of cashew nuts per hectare [4].

However, despite the potential of cashew production in the Northeast region, the irregularity and poor distribution of rainfall in the region, associated with high temperatures, results in high evapotranspiration rates, complicating the cultivation of this crop [5]. In this scenario, the employment of irrigation emerges as a viable solution for continuous cultivation, allowing the expansion of productive regions over the year [6]. However, most water sources (surface and subsurface) show high concentrations of dissolved salts, which constitutes a limiting factor for the cultivation of species sensitive to salt stress, e.g., the precocious-dwarf cashew. Excess salt in the water and/or soil can affect water availability for crops due to osmotic and ionic effects [7].

The use of saline water for irrigation negatively affects the metabolic and biochemical functions of plants, inhibiting their growth and capacity to perform photosynthesis and protein synthesis, interfering with enzymatic activity and chlorophyll synthesis. Furthermore, saline water also influences the flow of electrons, altering the functioning of photosystem II [8,9]. The high concentrations of ions such as Na^+^ and Cl^−^ in the roots cause a series of morphophysiological disorders due to the osmotic and ionic effects of stress caused by salinity, decreasing water and nutrient uptake [10,11].

The search for strategies capable of alleviating salt stress effects on plants and enabling irrigation with saline water in irrigated fruit farming is essential to obtain high crop yields with economic advantage. In this scenario, the foliar application of salicylic acid (SA) stands out as a viable strategy [12]. SA is a natural phenolic compound present in some physiological processes of plants, e.g., floral induction, stomatal opening and closing, ion absorption, photosynthesis, transpiration, and in the activation and catalysis of antioxidative enzymes and biosynthetic proteins, degrading reactive oxygen species (ROS) [13,14]. However, the effects of SA depend on the species and stage of crop development, in addition to the concentration and application method employed [14,15].

Studies conducted by Ekbic et al. [16] to evaluate the effects of salicylic acid application at concentrations of 6 and 9 mM in American grapevines irrigated with water of electrical conductivity up to 8.0 dS m^−1^ highlighted a reduction in the deleterious effects caused by salt stress on plant growth and development. In another study, Samadi et al. [17] observed that the foliar application of salicylic acid at a concentration of 100 µM mitigated the harmful effects of salinity in strawberry plants irrigated with 50 mM NaCl solution. 

Silva et al. [14] studied a *soursop* crop irrigated with water of electrical conductivity levels ranging from 0.8 to 4.0 dS m^−1^, and observed that the foliar application of salicylic acid at the concentration of 2.75 mM increased the relative growth rate in stem diameter.

The present study is based on the hypothesis that the foliar application of salicylic acid at adequate concentrations mitigates the deleterious effects caused by irrigation with saline water on the growth and physiology of the precocious-dwarf cashew, inducing plant tolerance to salt stress caused by the increase in the biosynthesis of photosynthetic pigments and the photochemical efficiency, reflecting on higher plant growth. From this perspective, this research aimed to evaluate the effects of the foliar application of salicylic acid on the chlorophyll fluorescence, photosynthetic pigments, and growth of precocious-dwarf cashew as a function of irrigation with saline water and the foliar application of salicylic acid during the post-grafting phase.

## 2. Results and Discussion

There was a significant effect of the interaction between irrigation water salinity levels and salicylic acid concentrations on the relative water content (RWC) and electrolyte leakage (% EL) in the leaf blade of precocious-dwarf cashew plants (Table 1).

For the relative water content (Figure 1A), the SA concentration of 1.1 mM mitigated the effects of salt stress up to the electrical conductivity of irrigation water (ECw) of 1.0 dS m^−1^, promoting an increase of 3.53% compared to plants subjected to the salinity of 0.4 dS m^−1^ without application of salicylic acid (0 mM). On the other hand, as the irrigation water salinity increased and the salicylic acid concentrations decreased, the RWC was reduced, achieving the minimum value (38.63%) in plants subjected to the ECw of 3.6 dS m^−1^ and the SA concentration of 3 mM. The RWC reduction as a function of the increase in water salinity can be explained by the osmotic effect, which restricts water uptake by plants and affects their water potential [18]. In a study evaluating the morphophysiology of the *soursop* cv. Morada Nova irrigated with saline water (ECw ranging from 0.8 to 4.0 dS m^−1^), Silva et al. [19] also observed a 13% reduction in the RWC as the salinity levels increased in the irrigation water.

For electrolyte leakage in the leaf blade (Figure 1B), the concentration of 1.1 mM of salicylic acid mitigated the effects of salt stress, achieving the lowest value of 38.63% in plants irrigated with water of 1.0 dS m^−1^. In plants subjected to the SA concentration of 0 mM and the ECw of 0.4 dS m^−1^, the % EL was 3.52% lower compared to plants grown under the concentration of SA 1.1 mM. The reduction in electrolyte leakage in the leaf blade of cashew plants can be explained by the protection of the cell membrane and the photosynthetic activity, as SA interacts with the signaling of ROS, reducing oxidative stress [20,21,22]. It should be noted that the electrolyte leakage observed in this study did not cause injuries to the leaf tissues since, according to Sullivan [23], the cell is considered injured when the damage percentage surpasses 50%. However, the increase in the ECw associated with salicylic acid concentrations higher than 1.1 mM intensified the effects of salt stress on cashew plants, increasing electrolyte leakage in the cell membrane, with the highest estimated value of 38.63% in plants irrigated with water of the electrical conductivity of 3.6 dS m^−1^ and foliar application of 3.0 mM of SA.

There was a significant effect of the irrigation water salinity on the contents of chlorophyll a (Chl *a*), b (Chl *b*), total chlorophyll (Chl *total*), and carotenoids (Car) of precocious-dwarf cashew plants at 280 DAT. The SA (0, 1, 2, and 3 mM) concentrations significantly influenced the contents of chlorophyll a and b (Table 2). The interaction between factors (S × SA) did not affect traits analyzed 280 days after transplanting.

The salinity levels of irrigation water influenced the contents of chlorophyll a (Figure 2A), achieving the maximum estimated value of 190.47 µg mL^−1^ in plants subjected to the ECw of 1.1 dS m^−1^. On the other hand, the minimum estimated value of 141.43 µg mL^−1^ was observed in plants grown under the ECw of 3.6 dS m^−1^. The reduction in the contents of chlorophyll a is a result of the increase in the water salinity levels, increasing the activity of the chlorophyllase enzyme and being related to the reduction in the number of chloroplasts, affecting the thylakoid membranes and constituting a recurrent symptom of oxidative stress [24,25].

In a study conducted by Lima et al. [26] using the precocious-dwarf cashew cv. Embrapa 51 irrigated with saline water (ECw ranging from 0.4 to 3.6 dS m^−1^), the authors observed a reduction of 17.10% in the Chl *a* content per unit increase in the ECw. The reduction in the chlorophyll contents is related to lipid peroxidation and the increase in the generation of ROS [27].

The concentrations of salicylic acid increased the chlorophyll *a* content (Figure 2B) up to the estimated concentration of 1.7 mM, whose maximum estimated value was 194.03 µg mL^−1^. Salicylic acid plays an important antioxidant role, increasing the activity of the peroxidase, superoxide dismutase, and catalase enzymes to eliminate reactive oxygen species, thus increasing chlorophyll synthesis [28,29]. Similar results were found by Silva et al. [30] in a study with *soursop* under salt stress (ECw ranging from 0.8 to 4.0 dS m^−1^), as the application of 1.4 mM of SA mitigated the effects of salt stress up to the ECw of 1.5 dS m^−1^. 

The chlorophyll *b* contents of precocious-dwarf cashew plants were reduced linearly with the increase in the irrigation water salinity (Figure 2C), amounting to an 11.5% per unit increase in the ECw. When comparing the Chl *b* contents of plants irrigated with water with the highest salinity (3.6 dS m^−1^) to those with the lowest salinity (0.4 dS m^−1^), there was a reduction of 38.77%. The inhibition in chlorophyll synthesis is related to the increase in the synthesis of 5-aminolevulinic acid, a molecule responsible for chlorophyll production and that acts in the degradation of photosynthetic pigment molecules [31].

The concentrations of salicylic acid also influenced the chlorophyll *b* contents of the precocious-dwarf cashew plants (Figure 2D), achieving the maximum estimated value of 94.61 µg mL^−1^ in plants without its application (0 mM). In turn, the minimum value of 62.11 µg mL^−1^ was achieved in plants subjected to the foliar application of SA 1 mM. The benefits of salicylic acid are related to the increase in enzymatic and photosynthetic activity, in addition to maintaining the balance between the degradation and production of reactive oxygen species [20,32]. 

The results found for chlorophyll *b* agree with the study conducted by Lima et al. [33], in which the authors analyzed three clones of precocious-dwarf cashew (Faga 11, Embrapa 51, CCP 76) irrigated with water of different electrical conductivity (0.4, 1.2, 2.0, 2.8, and 3.6 dS m^−1^), obtaining reductions of, respectively, 16.86, 14.86, and 16.92% per unit increase in the ECw. 

The carotenoid and total chlorophyll contents (Figure 3A,B) of the precocious-dwarf cashew plants decreased linearly with the increase in the electrical conductivity of irrigation water, with reductions of 7.55 and 8.42% per unit increase in the ECw. When comparing the carotenoid and total chlorophyll contents of plants irrigated with the highest salinity (ECw = 3.6 dS m^−1^) with those that received the ECw of 0.4 dS m^−1^, there was a reduction of 24.94 and 27.89%, respectively.

The reduction in the carotenoid contents of plants grown employing high salinity waters highlights the damage that occurred to the photosynthetic apparatus since carotenoids are involved in the transfer of captured light to chlorophyll, thus influencing this transfer and the photosynthetic relations [34]. Conversely, Lima et al. [26] evaluated precocious-dwarf cashew plants irrigated with ECw levels ranging from 0.4 to 3.6 dS m^−1^ and verified a 60.03% increase in the carotenoid contents of these plants.

There was a significant effect (*p* ≤ 0.05 and 0.01) of the electrical conductivity levels of irrigation water on the initial (F_0_), maximum (Fm), and variable fluorescence (Fv) and on the quantum efficiency of photosystem II (Fv/Fm) of precocious-dwarf cashew plants (Table 3), 280 days after transplanting.

Water salinity linearly increased the initial fluorescence of precocious-dwarf cashew plants (Figure 4A), with an increase of 8.29% per unit increase in the ECw. When comparing the F_0_ of plants irrigated with 3.6 dS m^−1^ to those under the ECw of 0.4 dS m^−1^, there was an increase of 21.19%. The F_0_ increase leads to lower utilization of available energy, which explains the damage caused by salt stress in the capture of light energy by photosynthetic pigments [35,36]. This behavior was already observed in cashew plants by Lima et al. [37], in a study where the authors observed a 19.68% increase in the F_0_ of plants irrigated with water of electrical conductivity up to 3.6 dS m^−1^ in relation to plants under the lowest water salinity (0.4 dS m^−1^). Silva et al. [30] evaluated the photosynthetic efficiency of the *soursop* (*Annona muricata* L.) cv. Morada Nova under salt stress (ECw ranging from 0.8 to 4.0 dS m^−1^) and observed that the increase in the ECw of irrigation water increased the initial fluorescence by 2.27% per unit increase in the ECw.

Conversely to what was observed in the F_0_ (Figure 4A), the maximum fluorescence decreased with the increase in the ECw (Figure 4B), reducing by 6.16% per unit increase in the ECw. The plants irrigated with the ECw of 3.6 dS m^−1^ showed an Fm reduction of 20.21% (263.57) compared to those cultivated with the lowest salinity (0.4 dS m^−1^). Salinity reduces the capture of energy in the reaction centers, probably because the excessive accumulation of specific ions causes an imbalance in the plant’s metabolic activity, leading to the formation of reactive oxygen species, which limits the energy activity of photosynthetic pigments [38].

Fernandes et al. [39] studied the effects of irrigation with saline water (ECw of 1.3 and 4.0 dS m^−1^) in custard apples and observed a reduction of 14.59% in the Fm of plants irrigated with the ECw of 4.0 dS m^−1^ in relation to those that received 1.3 dS m^−1^. Furthermore, according to these authors, the reduction in the maximum fluorescence can be explained by the action of the excess of salts in the photoreduction of quinone and in the thylakoid membranes, because of the entry of electrons into the photosystem. 

Similar to the Fm, the variable fluorescence (Fv) decreased with the increase in irrigation water salinity (Figure 4C), whose decrease was 10.77% per unit increase in the ECw. When comparing in relative terms, there was a 36.01% reduction in the Fv of plants irrigated with the ECw of 3.6 dS m^−1^ compared to those subjected to the water salinity of 0.4 dS m^−1^. Since it corresponds to the active potential energy in the photosystem, the Fv reduction demonstrates the limitation in the activation of the electron transport chain, which is responsible for the production of ATP and NDPH in the Calvin cycle, reducing the plant’s photosynthetic capacity [2,36]. 

Similar results were reported by Silva et al. [30] when analyzing the photochemical efficiency of the *soursop* (*Annona muricata* L.) cv. Morada Nova plants irrigated with saline water (ECw of 0.8 to 4.0 dS m^−1^), observed an Fv reduction of 17.41% in plants grown under the ECw of 4.0 dS m^−1^ in comparison to those that received 0.8 dS m^−1^. 

The quantum efficiency of photosystem II was also affected by salt stress (Figure 4D), showing a reduction of 5.81% per unit increase in the ECw, equivalent to a 19.05% reduction between plants cultivated under the water salinity levels of 0.4 and 3.6 dS m^−1^. This situation indicates photochemical damage to cashew plants under salt stress, with part of the light energy available in the thylakoid membrane associated with the metabolic damage of salt stress, accelerating the production of ROS and degrading chlorophylls in the reaction center [35,38]. Similar responses were observed in studies developed by Diniz et al. [40] in yellow passion fruit (*Passiflora edulis* f; *flavicarpa*) and by Xavier et al. [41] in guava (*Psidium guajava* L.). 

There was a significant effect of the interaction between water salinity (S) and the concentrations of SA on the stem diameter below the grafting point (SD_rootstock_) and the diameter above the grafting point (SD_scion_) (Table 4) of precocious-dwarf cashew plants. As an isolated factor, the salinity (S) significantly affected the diameter at the grafting point (SD_grafting point_) and the vegetative vigor index (VVI). While the concentrations of salicylic acid significantly influenced plant height (PH) and the vegetative vigor index.

The increase in water salinity reduced the stem diameter below and above the grafting point of precocious-dwarf cashew plants (Figure 5A,B), achieving the maximum values of 30.94 and 17.27 mm, respectively, at the ECw of 0.4 dS m^−1^ and the SA concentration of 3 mM. On the other hand, the minimum values of 23.72 and 13.87 mm were observed in plants subjected to the ECw of 3.6 dS m^−1^ and 0 mM of SA. The application of salicylic acid increased the growth in diameter below and above the grafting point under irrigation with saline water, highlighting the role of this plant hormone in the regulation of plant development processes, acting in root growth, meristem expansion, and in the gas exchange variables [19,42].

Water salinity reduced the growth in stem diameter at the grafting point (Figure 6A), whose maximum estimated value of 26.53 mm was observed in plants irrigated with the ECw of 0.4 dS m^−1^. On the other hand, a minimum value of 22.08 mm was observed in plants cultivated under the ECw of 3.6 dS m^−1^. The inhibition in stem diameter growth could be associated with limitations in water and nutrient uptake and the accumulation of toxic ions (Na^+^ and Cl^−^), limiting pectin as a function of calcium deficiency (Ca^2+^) and resulting in cellular disruption, decreasing the rigidity of the cell wall, and directly affecting cell expansion in the stem [43]. Similar results were obtained by Lacerda et al. [6] when studying the guava cv. Paluma under water salinity (ECw of 0.4 and 3.2 dS m^−1^), observing a 13.25% decrease in the diameter of the rootstock with an increase in water salinity 390 days after transplanting.

The height of precocious-dwarf cashew plants (Figure 6B) increased linearly with the increase in the salicylic acid concentrations, with a 3.78% per unit increase in the SA concentration. When comparing, in relative terms, the growth of plants under the SA concentration of 3.0 mM compared to those without application (0 mM), there was an increase of 9.79 cm in PH. Salicylic acid is an essential hormone for plant development as it plays multiple roles that promote plant growth. SA stimulates cell division, resulting in a significant increase in plant growth. Furthermore, this hormone also activates metabolic pathways involved in plant growth, thus boosting plant development more robustly and vigorously [44].

The vegetative vigor index (VVI) (Figure 7A) of precocious-dwarf cashew plants linearly decreased with the increase in the ECw, with a 2.86% reduction per unit increase in the ECw. When comparing the VVI of plants irrigated with the ECw of 3.6 dS m^−1^ with those grown under the ECw of 0.4 dS m^−1^, there is a reduction of 10.23%. The SA concentrations also influenced the vegetative vigor index linearly (Figure 7B), with an increase of 2.66% per unit increase in the SA. Plants subjected to application with an SA of 3.0 mM showed an increase of 9.72% in the VVI compared to those under treatment without its application (0 mM).

The VVI reduction due to the increase in water salinity can be explained by the osmotic stress in plants, which results in an imbalance in plants cells, leading to the reduction in water availability as the consequence of a negative impact in nutrient uptake by the roots, causing nutrient deficiency and decreasing cell turgidity [45]. These results agree with studies conducted by Lacerda et al. [6] in the guava cv. Paluma under irrigation with saline water (ECw of 0.6 and 3.2 dS m^−1^), in which the authors observed that the increase in the ECw levels reduced the VVI 390 days after transplanting.

The interaction between water salinity and salicylic acid concentrations significantly interfered in the relative growth rate of the rootstock (RGB_SDrootstock_) and the relative growth rate of the scion (RGB_SDscion_) of precocious-dwarf cashew plants during the period 220 to 280 DAT (Table 5).

The cashew plants irrigated with the ECw of 0.8 dS m^−1^ and subjected to the SA concentration of 1.0 mM obtained higher relative growth rates in both rootstock and scion diameter (Figure 8A,B), with 0.0057 and 0.0041 mm mm^−1^ day^−1^ for the RGB_Sdrootstock_ and RGB_Sdscion_, respectively. The cashew plants irrigated with the Ecw of 0.8 dS m^−1^ and subjected to the SA concentration of 1.0 mM increased by 5.65% (0.00031 mm mm^−1^ day^−1^) and 7.9% (0.0003 mm mm^−1^ day^−1^) the RGB_Sdrootstock_ and RGB_Sdscion_, respectively, in relation to those under the Ecw of 0.8 dS m^−1^ and without SA application (0 mM). Furthermore, the foliar application of salicylic acid at concentrations higher than 1.0 mM intensified the effects of salt stress, with the lowest RGB_Sdrootstock_ (0.0025 mm mm^−1^ day^−1^) and RGB_Sdscion_ (0.0023 mm mm^−1^ day^−1^) values recorded in plants irrigated with the Ecw of 3.6 dS m^−1^ and sprayed with SA at the concentration of 3.0 mM.

Salicylic acid is considered a signaling molecule and acts as a natural regulator of plant growth [46]. As verified in the present study, the beneficial effect of salicylic acid on early dwarf cashew plants depended on the applied concentration. In addition to concentration, the effect of salicylic acid is related to the stage of crop development, method, and frequency of application [15,47,48]. 

SA can act as a signaling molecule changing the expression of antioxidant genes and influencing the quantity and/or activity of proteins under salinity, enabling a greater accumulation of ions responsible for the osmoregulation and structuring of membranes, e.g., K^+^ and Ca^2+^, and reducing the concentration of toxic ions such as Na^+^ and Cl^−^ [11]. This fact could be related to the increase in the relative growth rate in diameter of the rootstock and scion observed in this study (Figure 8).

In general, the results obtained in the present study indicate that the saline stress caused by the increase in the electrical conductivity of the irrigation water increased the percentage of electrolyte leakage, reduced the relative water content in the leaves of the early dwarf cashew plant, and limited the synthesis of photosynthetic pigments and the quantum efficiency of photosystem II. This is directly reflected in the reduction in growth, especially in the control (Figure 9), i.e., in the plants that did not receive foliar application of salicylic acid (0 mM).

On the other hand, the foliar application of SA between concentrations of 1.0 and 1.1 mM attenuated the effects of salt stress on the relative water content of the leaf and electrolyte leakage. Furthermore, the 1.7 mM concentration of SA increased the synthesis of photosynthetic pigments. The growth variables of the dwarf-early cashew plant also increased as a function of the increase in salicylic acid concentrations (Figure 10). 

## 3. Materials and Methods

### 3.1. Location of the Experiment

The experiment was conducted from February to November 2022 under plant nursery conditions at the Agricultural Engineering Academic Unit of the Federal University of Campina Grande, PB, located at the local geographic coordinates 07°15′18″ S, 35°52′28″ W, and at a mean elevation of 550 m. The climate of the region is characterized, due to its milder temperatures, as tropical with dry summers (As), with a mean annual rainfall of 802.7 mm, maximum temperature of 27.5 °C, minimum temperature of 19 °C, and relative air humidity of 83% [49]. The data on the maximum and minimum temperatures and relative humidity of air observed during the experimental period are shown in Figure 11. 

### 3.2. Treatments and Experimental Design

A randomized block design was adopted in a 5 × 4 factorial arrangement whose treatments consisted of the combination of two factors: five electrical conductivity levels of irrigation water–ECw (0.4, 1.2, 2.0, 2.8, and 3.6 dS m^−1^) and four salicylic acid concentrations (0, 1, 2, and 3 mM), with three replications (Table 6). The distribution of treatments in the experimental area can be seen in Figure 12. The electrical conductivity levels were established according to the methodology described by Lima et al. [33] for precocious-dwarf cashew, with ECw levels from 0.4 dS m^−1^ inhibiting the growth of precocious-dwarf cashew plants during rootstock formation; while concentrations of salicylic acid were based on the study realized by Silva et al. [19] with the *soursop* cv. Morada nova.

### 3.3. Experiment Setup and Conduction 

The seedlings were acquired from a commercial orchard registered with the National Registry for Seeds and Seedlings, located in the municipality of Pacajus–CE, and grown in 0.5-L polyethylene bags measuring 10 × 20 cm. The seedlings were cleft-grafted onto the rootstocks, i.e., the cashew clones CCP 76 and BRS 226 Planalto.

Clone CCP 76 is recommended for cultivation under dryland and irrigated conditions, standing out for the utilization of the peduncle for the fresh consumption market, with one of the best flavors available and large-scale production. In turn, the clone BRS 226 Planalto shows high yields and high peduncle and nut quality, with resistance to resinosis and stem rot, diseases that have caused considerable losses to cashew producers [50].

Containers adapted as drainage lysimeters with a capacity of 250 L were used in the experiment. The lysimeters were perforated at the base and coupled to a transparent drain 20 mm in diameter to allow drainage. The end of the drain inside the lysimeters was wrapped with non-woven geotextile fabric (Bidim OP 30) to prevent obstruction with soil material. A plastic bottle was placed below each lysimeter to collect the drained water and estimate water consumption by the plant. 

The lysimeters were filled with a 0.5 kg layer of gravel, followed by 260 kg of previously ground Entisol with a sandy loam texture. The soil (0–30 cm depth–A horizon) was collected in the municipality of Riachão do Bacamarte, PB (07°15′34″ S, 35°40′01″ W, and elevation of 192 m). After collection and preparation, the soil was analyzed for physical and chemical attributes, according to the methodology of Teixeira et al. [51]: Ca^2+^ = 1.77 cmol_c_ kg^−1^; Mg^2+^ = 1.60 cmol_c_ kg^−1^; Na^+^= 0.51 cmol_c_ kg^−1^; K^+^= 0.20 cmol_c_ kg^−1^; H^+^+ Al^3+^= 3.15 cmol_c_ kg^−1^; organic matter = 9.30 dag kg^−1^; P = 10.70 dag kg^−1^; pH in water (1:2.5) = 4.93; electrical conductivity of saturation extract = 1.15 dS m^−1^; exchangeable sodium percentage = 7.05; sand = 760.6 g kg^−1^; silt = 164.5 g kg^−1^; clay = 74.9 g kg^−1^; moisture content at 33.42 kPa = 13.07 dag kg^−1^; moisture content at 1519.5 kPa = 5.26 dag kg^−1^; total porosity = 0.57 m^3^ m^−3^; bulk density = 1.13 kg dm^−3^; particle density = 2.65 kg dm^−3^.

The different electrical conductivity levels of irrigation water were prepared by dissolving NaCl, CaCl_2_.2H_2_O, and MgCl_2_.6H_2_O at an equivalent ratio of 7:2:1 between Na:Ca:Mg, respectively, in local tap water (ECw = 0.4 dS m^−1^). These salt proportions are commonly found in water sources used for irrigation in small properties in northeastern Brazil [52]. In the preparation of irrigation water, the relationship between the ECw and the concentration of salts [53] was considered, according to Equation (1):(1)C≈ 10 × ECw 
where:

C = Salt concentration (mmol_c_ L^−1^); 

ECw = Electrical conductivity of the water (dS m^−1^). 

After dissolving the NaCl, CaCl_2_.2H_2_O, and MgCl_2_.6H_2_O salts in water, calibration was performed using a bench conductivity meter (model DM-32, Digimed^®^, São Paulo, SP, Brazil), then the water was stored in water tanks with a capacity of 500 L.

Transplanting to the lysimeters was performed after opening holes measuring 20 × 20 × 20 cm. Subsequently, irrigation was performed daily using water with the lowest electrical conductivity (0.4 dS m^−1^). At 45 days after transplanting, irrigation with the different salinity levels was performed with a watering schedule of two days, and the volume of water applied was determined according to the water requirement of the plants, estimated by the water balance, as presented in Equation (2):(2)$$\mathrm{VI}=\frac{\left(\mathrm{Va}\;-\;\mathrm{Vd}\right)}{\left(1\;-\;\mathrm{LF}\right)}$$
where: 

VI—water volume to be used in the next irrigation event (mL);

Va—volume applied in the previous irrigation event (mL); 

Vd—drained volume (mL);

FL—0.15 leaching fraction, applied every 30 days.

The salicylic acid concentrations were obtained by dissolving salt in 30% ethyl alcohol, as SA is a substance of low solubility in water at ambient temperature. To reduce the superficial tension on the leaf surface, the Wil fix adjuvant was used in the preparation of the solution at the concentration of 0.5 mL L^−1^. The foliar applications began 30 days after transplanting (DAT), and then in 30-day intervals using a manual sprayer, from 5:00 to 6:00 p.m. The 12-L Jacto—Jacto XP sprayer was used for foliar application with a working pressure (maximum) of 88 psi (6 bar) and a JD 12P nozzle. During this period, an average volume of 194 mL of solution was applied per plant (Figure 13).

Fertilization with NPK was performed according to the recommendations of Oliveira [54] for the irrigated cultivation of precocious-dwarf cashew. Topdressing fertilization consisted of 60 g of N, 200 g of P_2_O_5_, and 40 g of K_2_O per plant per year, split into 24 applications at 15-day intervals. The nitrogen sources used were calcium nitrate and monoammonium phosphate; the phosphorus was applied through monoammonium phosphate, and the potassium source used was potassium sulfate. 

Fortnightly, a micro Dripsol^®^ solution was used to supply the micronutrient requirements at the concentration of 1.0 g L^−1^, with the following composition: Mg (1.1%), Zn (4.2%), B (0.85%), Fe (3.4%), Mn (3.2%), Cu (0.5%), and Mo (0.05%), via foliar application using a backpack sprayer. During the experiment, crop management practices such as manual weed removal, soil scarification, and pest and disease control were performed according to the requirements for the crop, the latter using deltamethrin at the concentration of 0.5 mL L^−1^ and azoxystrobin and difenoconazole at the concentration of 0.3 mL L^−1^.

### 3.4. Traits Analyzed

The following variables were evaluated 280 days after transplanting (DAT): relative water content (RWC), electrolyte leakage (% EL), photosynthetic pigments, chlorophyll fluorescence, and growth of precocious-dwarf cashew. For the determination of the relative water content (RWC), two leaves were removed from the middle third portion of the main branch to obtain five disks which were 12 mm in diameter. Immediately after collection, the disks were weighed to prevent loss of moisture, thus obtaining the fresh mass (FM). Next, these samples were placed in a beaker, immersed in 50 mL of distilled water, and stored for 24 h. After this period, the excess water from the disks was removed with paper towels, and the turgid mass (TM) of the samples was obtained. Next, the samples were dried to constant weight in a forced-air oven at 65 ± 3 °C to obtain the dry mass (DM) of the samples. The RWC was determined according to Weatherley [55], using Equation (3):(3)$$\;\mathrm{RWC}\;=\frac{\mathrm{FM}\;-\mathrm{TM}}{\mathrm{TM}\;-\;\mathrm{DM}}\times 100$$
where:

RWC = relative water content (%);

FM = fresh leaf mass (g);

TM = turgid mass of leaf (g); 

DM = dry mass of leaf (g).

Electrolyte leakage (% EL) in the leaf blade was determined using a copper perforator to obtain five disks of 1.54 cm^2^ per experimental unit, which were washed and placed in Erlenmeyer^®^ flasks containing 50 mL of distilled water. After being closed with aluminum foil, the Erlenmeyer^®^ flasks were exposed to ambient temperature at 25 °C for 24 h, after which the initial electrical conductivity of the medium (Xi) was determined using a benchtop conductivity meter (MB11, MS Techonopon^®^, Piracicaba, SP, Brazil). Next, the Erlenmeyer^®^ flasks were exposed to a temperature of 90 °C for 120 min in a drying oven (SL100/336, SOLAB^®^, Piracicaba, SP, Brazil). Then, after the material was cooled, the final electrical conductivity was measured (Xf). The electrolyte leakage in the leaf blade was expressed as the percentage of initial electrical conductivity in comparison to the electrical conductivity after 120 min of treatment at 90º C, according to the methodology proposed by Scotti-Campos et al. [56], considering Equation (4):(4)$$\%\;\mathrm{EL}=\frac{\mathrm{Xi}}{\mathrm{Xf}}\times 100$$
where:

EL—electrolyte leakage (%);

Xi—initial electrical conductivity (dS m^−1^); 

Xf—final electrical conductivity (dS m^−1^).

The contents of photosynthetic pigments (chlorophyll *a*, *b*, *total*, and carotenoids) were determined according to Arnon [57] using plant extracts collected from leaf disks from the third mature leaf counting from the apex. Each sample received 6.0 mL of 80% P.A. acetone. These extracts were used to determine the chlorophyll and carotenoid concentrations in the solutions using a spectrophotometer (model UV/VIS-UV1720, AKSON^®^, São Leopoldo, RS, Brazil) at the absorbance wavelength (ABS) (470, 647, and 663 nm) using Equations (5)–(8):
(5)Chl *a* = (12.25 × ABS663) − (2.79 × ABS647)
(6)Chl *b* = (21.50 × ABS647) − (5.10 × ABS647)
(7)Chl *total* = (7.15 × ABS663) + (18.71 × ABS647)
(8)$$\mathrm{Car}\;=\frac{\left|\left(1000\times \mathrm{ABS}470\right)-\left(1.82\times \mathrm{Chl}\;a\right)-\left(85.02\times \mathrm{Chl}\;b\right)\right|}{198}$$

The values obtained for the contents of chlorophyll *a*, *b*, *total*, and carotenoids in the leaves were expressed as µg mL^−1^.

Chlorophyll fluorescence was determined using a modulated pulse fluorometer, model OS5p by Opti Science. The variables evaluated were the initial fluorescence (F_0_), maximum fluorescence (Fm), variable fluorescence (Fv), and the quantum efficiency of photosystem II (Fv/Fm). The Fv/Fm protocol was performed after the leaves were adapted to the dark for 30 min using a clip of the equipment to ensure that all acceptors were oxidized, i.e., the reaction centers were opened.

Plant height was measured by considering the distance from the base of the plant to the insertion of the apical meristem; stem diameter consisted of measurements below the grafting point (SD_rootstock_), at the grafting point (SD_grafting point_), and above the grafting point (SD_scion_) using a digital caliper. The crown diameter (D_Crown_) was determined by the average crown diameter in the row (DR) and the interrow direction (DIR). The canopy volume (V_Crown_) and the vegetative vigor index (VVI) were calculated according to Portella et al. [58] using Equations (9) and (10):(9)$$V_{\mathrm{Crown}}=\frac{\pi }{6}\times H_{\mathrm{Crown}}\times \mathrm{DR}\times \mathrm{DIR}$$
(10)$$VVI=\frac{\left[H_{Crown}+D_{Crown}+\left(SD\times 10\right)\right]}{100}$$
where:

V_Crown_—crown volume (m^3^);

VVI—vegetative vigor index;

H_Crown_—crown height (m);

DR—crown diameter in the row direction (m);

DIR—crown diameter in the interrow direction (m);

SD—stem diameter (mm).

The relative growth rates of the rootstock (RGB_SDrootstock_) and scion (RGB_SDscion_) were determined during the period from 220 to 280 DAT using data from the rootstock and scion according to the methodology described by Benincasa [59], using Equation (11).
(11)$$\mathrm{RGB}\;=\frac{\left(\mathrm{Ln}A_{2}-\mathrm{Ln}A_{1}\right)}{\left(T2\;-T1\right)}$$
where:

RGB—relative growth rate (mm mm^−1^ day^−1^);

A1—variable at time T1; 

A2—variable at time T2; 

T1—time 1 in days; 

T2—time 2 in days;

### 3.5. Statistical Analysis

The data obtained were tested for the normality of the distribution (Shapiro-Wilk test) at a 0.05 probability, followed by an analysis of variance by the F-test at 0.05 and 0.01 probability. When significant effects occurred, linear and quadratic polynomial regressions were performed for the salinity levels of irrigation water and concentrations of salicylic acid using the statistical software SISVAR-ESAL version 5.6 [60]. For the interaction between water salinity and salicylic acid concentrations (S × SA), the software SigmaPlot v. 14.5 was used to construct the graphs of the response surfaces.

## 4. Conclusions

Irrigation water with an electrical conductivity level above 0.4 dS m^−1^ negatively affects the relative water content in the leaf blade, the synthesis of photosynthetic pigments, the fluorescence of chlorophyll *a*, and plant growth, while increasing electrolyte leakage in the leaf blade of precocious-dwarf cashew plants in the absence of salicylic acid. The foliar application of salicylic acid at concentrations between 1.0 and 1.1 mM attenuates the effects of salt stress on the relative water content, electrolyte leakage in the leaf blade of precocious-dwarf cashew plants, and the relative growth rate in stem diameter of the rootstock and scion. In addition, the concentration of 1.7 mM of salicylic acid increases the synthesis of photosynthetic pigments in precocious-dwarf cashew plants 280 days after transplanting. These results reinforce the hypothesis that foliar application of salicylic acid at adequate concentrations can act as a signaling molecule and reduce the effects of salt stress in cashew plants, which can facilitate the use of brackish water in agricultural activity, especially in semi-arid regions of Northeastern Brazil. However, further studies are needed to elucidate the mechanisms of action of salicylic acid in salt stress signaling, in addition to identifying the proper frequency and/or form of application.

## Figures and Tables

**Figure 1 plants-12-02783-f001:**
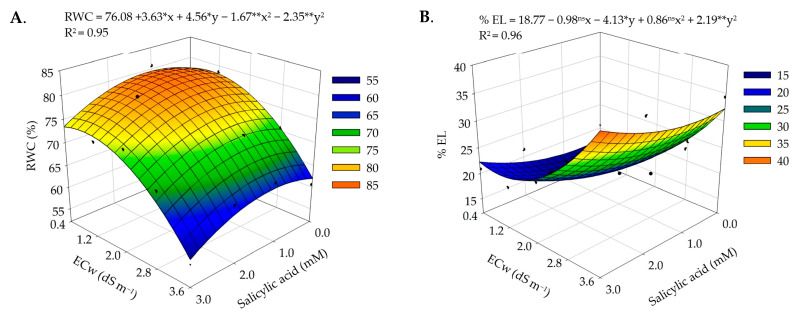
Relative water content—RWC (**A**) and electrolyte leakage–% EL (**B**) in the leaf blade of precocious-dwarf cashew plants as a function of the interaction between the electrical conductivity of irrigation water–ECw and salicylic acid concentrations–SA 280 days after transplanting. X and Y–Concentrations of SA and ECw, respectively; *, **, and ns = significant at a *p* ≤ 0.05, *p* ≤ 0.01, and non-significant, respectively.

**Figure 2 plants-12-02783-f002:**
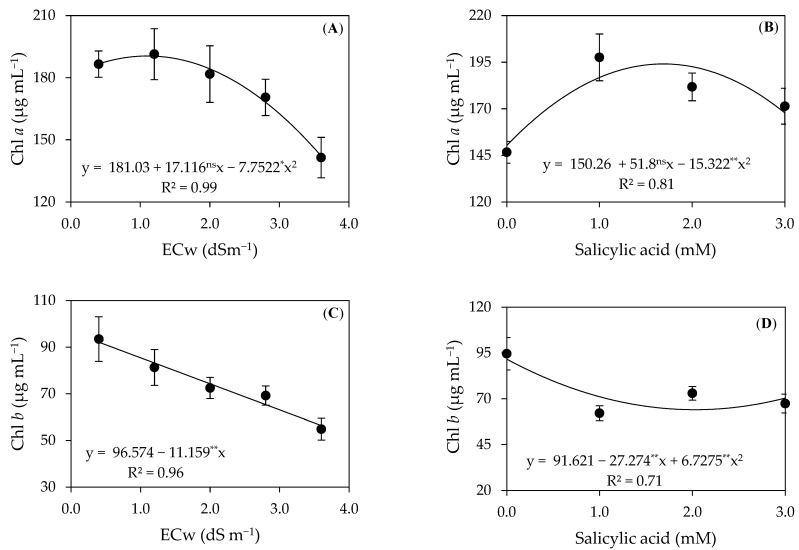
Chlorophyll *a* (Chl *a*) and chlorophyll *b* (Chl *b*) contents of precocious-dwarf cashew plants as a function of the electrical conductivity of irrigation water–ECw (**A**,**C**) and salicylic acid concentrations (**B**,**D**) 280 days after transplanting. *, **, and ^ns^: Significant at a *p* ≤ 0.05, 0.01, and non-significant, respectively. Vertical bars represent the standard error of the mean (*n* = 3).

**Figure 3 plants-12-02783-f003:**
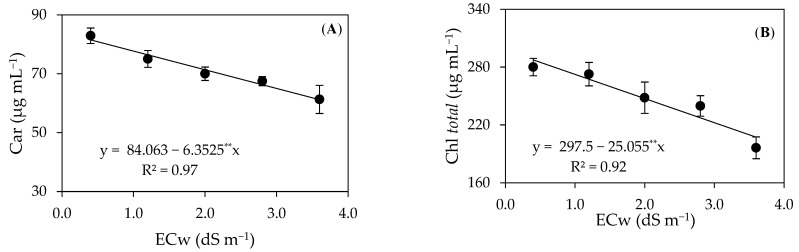
Contents of carotenoids–Car (**A**) and total chlorophyll–Chl *total* (**B**) of precocious-dwarf cashew plants as a function of the electrical conductivity of irrigation water–ECw 280 days after transplanting. ** Significant at a *p* ≤ 0.01. Vertical bars represent the standard error of the mean (*n* = 3).

**Figure 4 plants-12-02783-f004:**
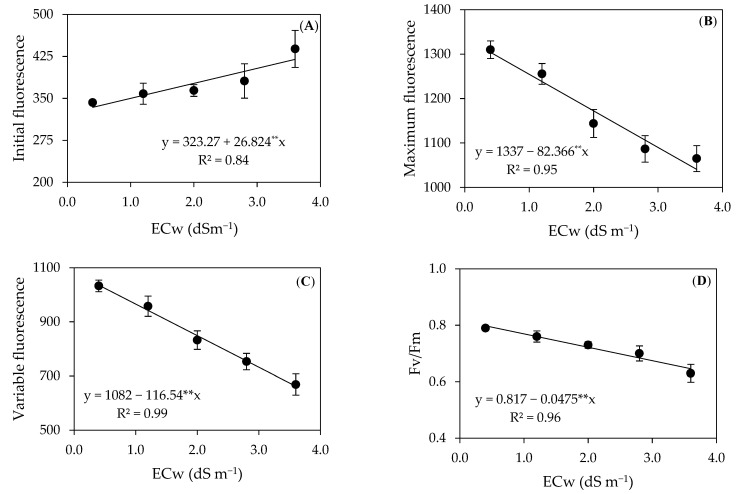
Initial–F_0_ (**A**), maximum–Fm (**B**), and variable fluorescence–Fv (**C**), and photochemical efficiency of photosystem II–Fv/Fm (**D**) of precocious-dwarf cashew plants as a function of the electrical conductivity of irrigation water–ECw 280 days after transplanting. ** Significant at a *p* ≤ 0.01 by the F-test. Vertical bars represent the standard error of the mean (*n* = 3).

**Figure 5 plants-12-02783-f005:**
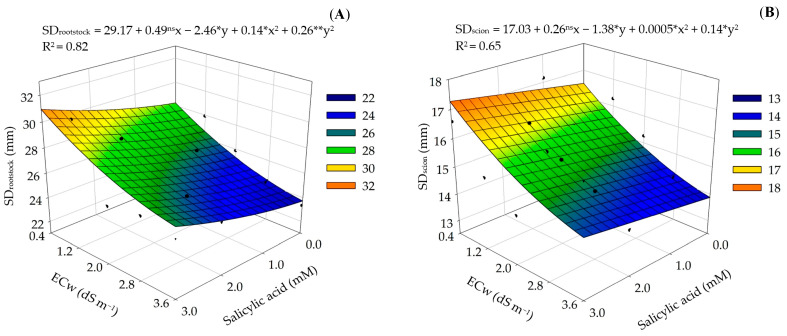
Diameter below the grafting point–SD_rootstock_ (**A**) and above the grafting point–SD_scion_ (**B**) of precocious-dwarf cashew plants as a function of the interaction between the electrical conductivity levels of irrigation water—ECw and salicylic acid concentrations–SA 280 days after transplanting. X and Y–Concentrations of SA and ECw, respectively; *, **, and ns: Significant at a *p* ≤ 0.05, 0.01, and non-significant, respectively.

**Figure 6 plants-12-02783-f006:**
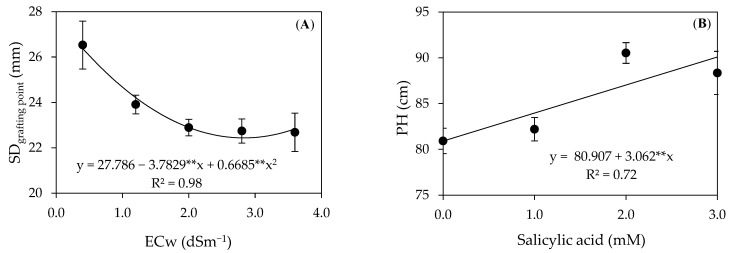
Diameter at the grafting point–SD_grafting point_ (**A**) and height–PH (**B**) of precocious-dwarf cashew plants as a function of irrigation water salinity–ECw and salicylic acid concentrations 280 days after transplanting. ** Significant at a *p* ≤ 0.01. Vertical bars represent the standard error of the mean (*n* = 3).

**Figure 7 plants-12-02783-f007:**
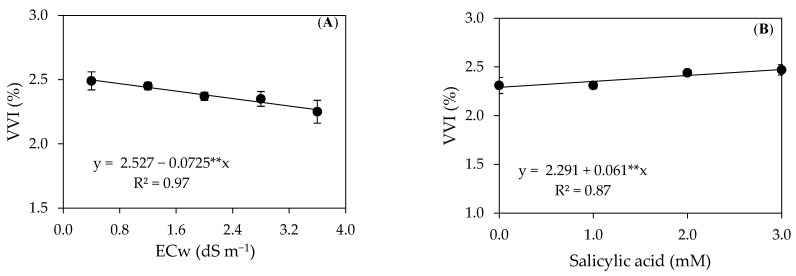
Vegetative vigor index–VVI as a function of irrigation water salinity (**A**) and the foliar application of salicylic acid–SA (**B**) of precocious-dwarf cashew plants 280 days after transplanting. ** Significant at a *p* ≤ 0.01. Vertical bars represent the standard error of the mean (*n* = 3).

**Figure 8 plants-12-02783-f008:**
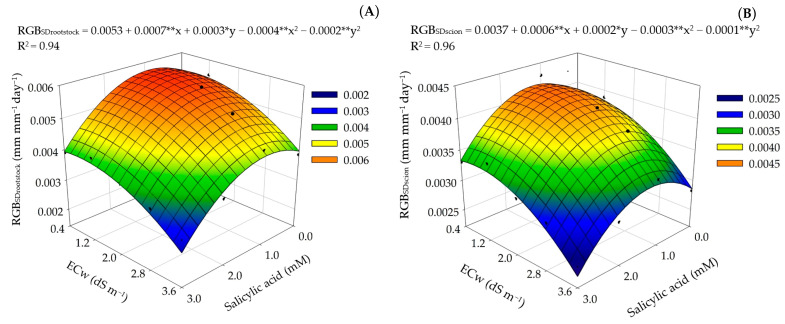
Relative growth rate in stem diameter of the rootstock—RGB_Sdrootstock_ (**A**) and scion—RGB_Sdscion_ (**B**) of precocious-dwarf cashew plants as a function of the interaction between the electrical conductivity levels of the irrigation water–Ecw and salicylic acid concentrations in the period from 220 to 280 days after transplanting. X and Y–concentration of SA and Ecw, respectively; * and ** significant at *p* ≤ 0.05 and 0.01, respectively.

**Figure 9 plants-12-02783-f009:**
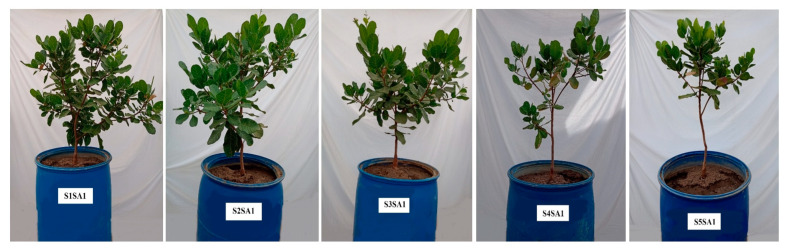
Comparison of early dwarf cashew plant morphology without foliar application of salicylic acid as a function of electrical conductivity of irrigation water. S = water salinity levels, SA = salicylic acid, S1 = 0.4 dS m^−1^, S2 = 1.2 dS m^−1^, S3 = 2.0 dS m^−1^, S4 = 2.8 dS m^−1^, S5 = 3.6 dS m^−1^, SA1 = 0 mM.

**Figure 10 plants-12-02783-f010:**
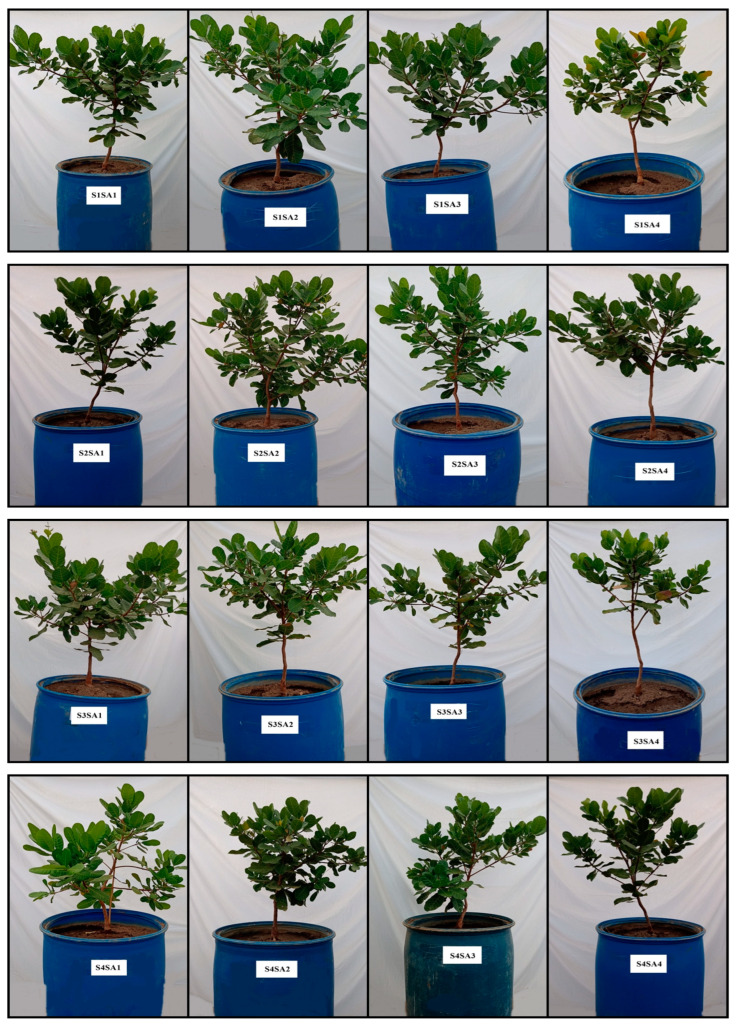
Comparison of early dwarf cashew plant morphology as a function of electrical conductivity of irrigation water and salicylic acid concentrations. S = water salinity levels, SA = salicylic acid, S1 = 0.4 dS m^−1^, S2 = 1.2 dS m^−1^, S3 = 2.0 dS m^−1^, S4 = 2.8 dS m^−1^, S5 = 3.6 dS m^−1^, SA1 = 0 mM, SA2 = 1 mM, SA3 = 2 mM, SA4 = 3 mM.

**Figure 11 plants-12-02783-f011:**
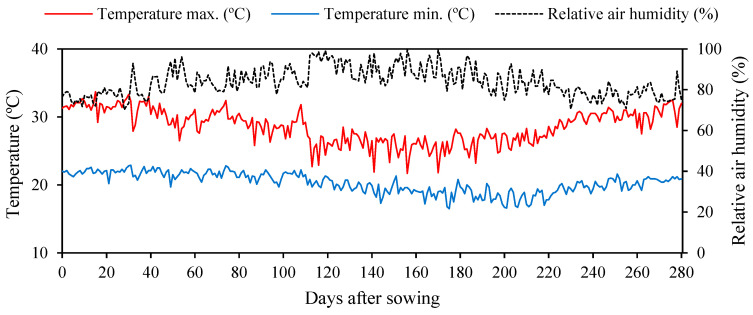
Meteorological data collected during the experimental period.

**Figure 12 plants-12-02783-f012:**
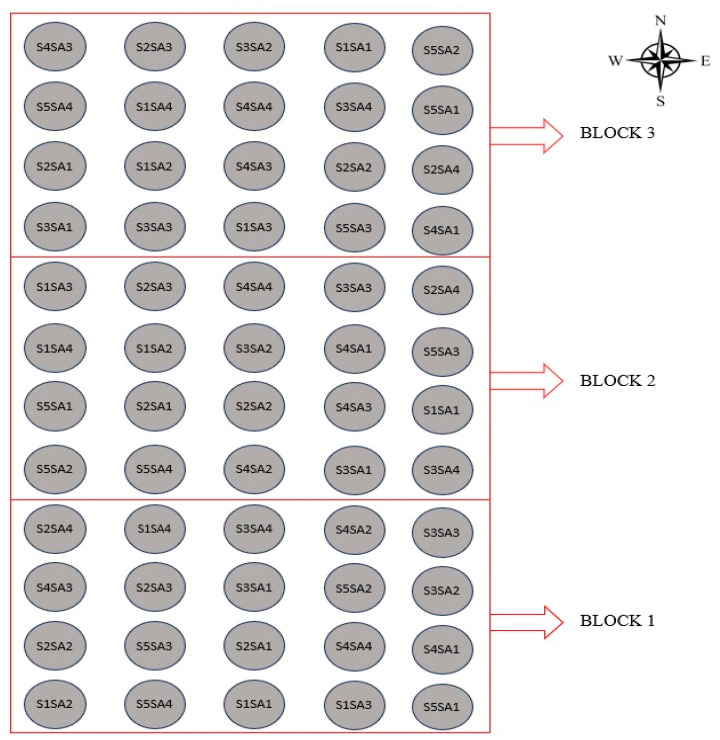
Distribution of treatments in the experimental area. S1 = 0.4 dS m^−1^, S2 = 1.2 dS m^−1^, S3 = 2.0 dS m^−1^, S4 = 2.8 dS m^−1^, S5 = 3.6 dS m^−1^, SA = salicylic acid, SA1 = 0 mM, SA2 = 1 mM, SA3 = 2 mM, SA4 = 3 mM.

**Figure 13 plants-12-02783-f013:**
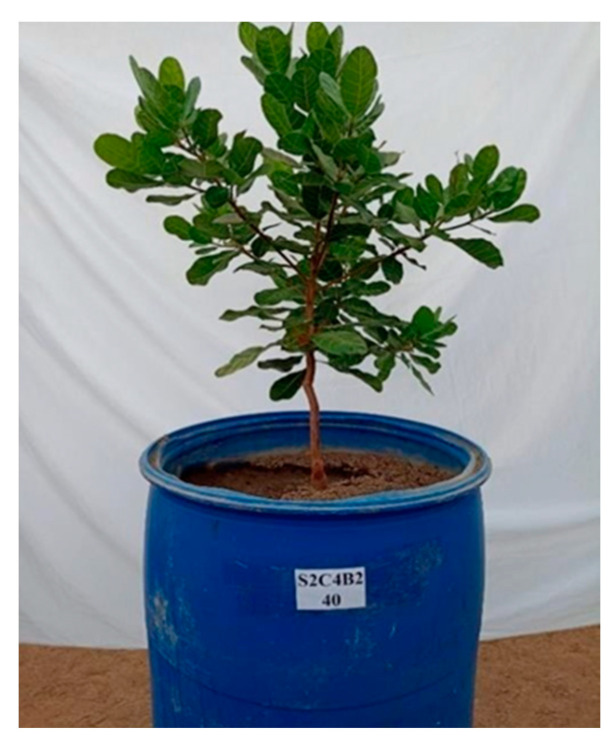
General aspect of cashew plant isolated by a plastic curtain to prevent drift during the application of salicylic acid.

**Table 1 plants-12-02783-t001:** Summary of the analysis of variance referring to the relative water content (RWC) and electrolyte leakage (% EL) in the leaf blade of precocious-dwarf cashew irrigated with saline water and foliar application of salicylic acid 280 days after transplanting.

Source of Variation	DF	Mean Squares	
		RWC	% EL
Salinity (S)	4	560.97 **	495.86 **
Linear regression	1	1843.26 **	1641.91 **
Quadratic regression	1	364.88 **	329.44 **
Salicylic acid (SA)	3	109.17 **	110.16 **
Linear regression	1	131.59 **	191.02 **
Quadratic regression	1	159.12 **	44.29 **
Interaction (S × SA)	12	14.01 **	7.59 *
Blocks	8	10.54 ^ns^	6.73 ^ns^
Residual	3	3.81	3.67
CV (%)		2.70	8.12

^ns^, *, **, respectively, non-significant and significant at a *p* ≤ 0.05 and *p* ≤ 0.01. CV: Coefficient of variation; DF: Degree of freedom.

**Table 2 plants-12-02783-t002:** Summary of the analysis of variance referring to the contents of chlorophyll a (Chl *a*), b (Chl *b*), carotenoids (Car), and total chlorophyll (Chl *total*) of precocious-dwarf cashew irrigated with saline water and foliar application of salicylic acid 280 days after transplanting.

Source of Variation	DF	Mean Squares			
		Chl *a*	Chl *b*	Car	Chl *Total*
Salinity (S)	4	4779.97 **	2468.87 *	796.35 **	13,243.18 **
Linear regression	1	14,828.74 **	9562.95 **	3099.47 **	48,208.62 **
Quadratic regression	1	4134.65 *	1.2411 ^ns^	29.48 *	3993.30 ^ns^
Salicylic acid (SA)	3	6849.98 **	3055.15 **	115.81 ^ns^	1556.24 ^ns^
Linear regression	1	2552.66 ^ns^	3771.73 **	62.62 ^ns^	118.64 ^ns^
Quadratic regression	1	140,487.66 **	2715.82 **	272.85 ^ns^	4433.81 ^ns^
Interaction (S × SA)	12	1097.39 ^ns^	498.76 ^ns^	147.17 ^ns^	1977.88 ^ns^
Blocks	8	2029.53 ^ns^	865.03 ^ns^	198.14 ^ns^	5377.52 ^ns^
Residual	3	918.14	289.94	91.70	1523.33
CV (%)		17.38	22.73	13.42	15.70

^ns^, *, **, respectively, non-significant and significant at a *p* ≤ 0.05 and *p* ≤ 0.01. CV: Coefficient of variation; DF: Degree of freedom.

**Table 3 plants-12-02783-t003:** Summary of the analysis of variance for the initial (F_0_), maximum (Fm), and variable fluorescence (Fv), and quantum efficiency of photosystem II (Fv/Fm) of precocious-dwarf cashew plants cultivated with saline water and concentrations of salicylic acid 280 days after transplanting.

Mean Squares					
Source of Variation	DF	F_0_	Fm	Fv	Fv/Fm
Salinity (S)	4	16,496.041 *	136,642.79 **	262,471.097 **	0.04619 **
Linear regression	1	5526.20 **	521,027.81 **	1,043,077.75 **	0.177 **
Quadratic regression	1	7668.00 ^ns^	12,325.89 ^ns^	627.21 ^ns^	0.00526 ^ns^
Salicylic acid (SA)	3	2189.75 ^ns^	3753.62 ^ns^	5578.44 ^ns^	0.002156 ^ns^
Linear regression	1	2852.08 ^ns^	4004.71 ^ns^	15,513.28 ^ns^	0.00563 ^ns^
Quadratic regression	1	350.41 ^ns^	245.19 ^ns^	1075.69 ^ns^	0.001 ^ns^
Interaction (S × SA)	12	6749.22 ^ns^	1564.98 ^ns^	9407.10 ^ns^	0.00668 ^ns^
Blocks	8	5273.45 ^ns^	34,893.38 ^ns^	49,964.031 ^ns^	0.00862 ^ns^
Residual	3	6180.82	9491.58	1035.32	0.005102
CV (%)		20.86	8.31	12.92	9.94

^ns^, *, **, respectively, non-significant, significant at *a p* ≤ 0.05, and *p* ≤ 0.01. CV: Coefficient of variation; DF: Degree of freedom.

**Table 4 plants-12-02783-t004:** Summary of the analysis of variance for the diameter below the grafting point (SD_rootstock_), above the grafting point (SD_scion_), and at the grafting point (SD_grafting point_), plant height (PH), canopy volume (V_Crown_), canopy diameter (D_Crown_), and vegetative vigor index (VVI) of precocious-dwarf cashew plants irrigated with saline water and foliar application of salicylic acid 280 days after transplanting.

Source of Variation	DF	Mean Squares						
		SD_rootstock_	SD_scion_	SD_grafting point_	PH	V_Crown_	D_Crown_	VVI
Salinity (S)	4	41.62 **	8.10 **	17.69 **	101.4 ^ns^	0.00252 ^ns^	0.00475 ^ns^	0.107 **
Linear regression	1	152.46 **	30.9 **	56.73 **	74.30 ^ns^	0.00917 ^ns^	0.00179 ^ns^	0.414 **
Quadratic regression	1	4.82 **	0.0572 ^ns^	13.57 **	0.43 ^ns^	0.000213 ^ns^	0.00047 ^ns^	0.00095 ^ns^
Salicylic acid (SA)	3	20.85 **	3.15 ^ns^	6.35 ^ns^	326.55**	0.00193 ^ns^	0.00318 ^ns^	0.114 **
Linear regression	1	61.32 **	7.76 ^ns^	0.139 ^ns^	703.52 **	0.0011 ^ns^	0.00042 ^ns^	0.293 **
Quadratic regression	1	1.14 ^ns^	1.4 ^ns^	0.2018 ^ns^	44.63 ^ns^	0.00016 ^ns^	0.00876 ^ns^	0.0026 **
Interaction (S × SA)	12	8.43 **	7.7 **	19.01 ^ns^	128.07 ^ns^	0.00851 ^ns^	0.0127 ^ns^	0.102 ^ns^
Blocks	8	0.34 ^ns^	0.85 ^ns^	0.011 ^ns^	38.66 ^ns^	0.00993 ^ns^	0.00913 ^ns^	0.0201 ^ns^
Residual	3	1.39	1.54	1.87	12.88	0.00193	0.00913	0.0197
CV (%)		4.40	8.14	5.81	4.20	19.28	6.87	5.89

^ns^, **, respectively, non-significant and significant at a *p* ≤ 0.01. CV: Coefficient of variation; DF: Degree of freedom.

**Table 5 plants-12-02783-t005:** Summary of the analysis of variance for the relative growth rate of the rootstock (RGB_SDrootstock_) and relative growth rate of the scion (RGB_SDscion_) of precocious-dwarf cashew plants irrigated under saline water and foliar application of salicylic acid during the period 220 to 280 days after transplanting.

Source of Variation	DF	Mean Squares	
		RGB_SDrootstock_	RGB_SDscion_
Salinity (S)	4	9.0 × 10^−6^ **	2.0 × 10^−6^ **
Linear regression	1	3.0 × 10^−5^ **	3.0 × 10^−6^ **
Quadratic regression	1	4.0 × 10^−6 ns^	2.0 × 10^−6^ **
Salicylic acid (SA)	3	5.0 × 10^−6^ **	4.0 × 10^−6^ **
Linear regression	1	1.1 × 10^−5^ *	1.0 × 10^−8 ns^
Quadratic regression	1	2.0 × 10^−6^ **	1.1 × 10^−6^ **
Interaction (S × SA)	12	3.0 × 10^−6^ **	2.0 × 10^−6^ **
Blocks	8	2.0 × 10^−6 ns^	2.5 × 10^−7 ns^
Residual	3	3.69 × 10^−7^	3.8 × 10^−7^
CV (%)		15.27	18.22

^ns^, *, **, respectively, non-significant, significant at *p* ≤ 0.05 and *p* ≤ 0.01. CV: Coefficient of variation; DF: Degree of freedom.

**Table 6 plants-12-02783-t006:** Description of the analyzed treatments.

ECw (S)	Salicylic Acid (SA) Concentrations			
	0 mM	1 mM	2 mM	3 mM
0.4 dS m^−1^	S1SA1 (control)	S1SA2	S1SA3	S1SA4
1.2 dS m^−1^	S2SA1	S2SA2	S2SA3	S2SA4
2.0 dS m^−1^	S3SA1	S3SA2	S3SA3	S3SA4
2.8 dS m^−1^	S4SA1	S4SA2	S4SA3	S4SA4
3.6 dS m^−1^	S5SA1	S5SA2	S5SA3	S5SA4

## Data Availability

Not applicable.

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
