# Peer review of "Salicylic Acid as a Salt Stress Mitigator on Chlorophyll Fluorescence, Photosynthetic Pigments, and Growth of Precocious-Dwarf Cashew in the Post-Grafting Phase"

_plants, 2023, doi:10.3390/plants12152783_

Round 1

Reviewer 1 Report

The manuscript reports the positive influence of salicylic acid in attenuating salt stress effects on some physiological attributes. Salicylic acid is now recognized as a potential plant growth regulator for promotion of photosynthesis and metabolites production and resistance to abiotic stress factors. Authors have justified the study by selecting the appropriate concentration of salicylic acid and salt stress. The manuscript, in particular has emphasized the positive role of salicylic acid through proper regression analyses. The presentation of the manuscript can be improved by simplifying the text and expression of results.  Following points may be considered for improvement.

· Please give more explanation for the clarity of the experiment, and mention the hierarchy of the treatments.

· Mention in the abstract that the electrical conductivity itself is the salt source (mention if otherwise).

· Clearly state the control setup for each experiment and ensure uniformity in the result.

· The results should be free from ambiguity and try to simplify it, easier to understand to those who are not adept in statistics. Try to emphasize the impact of salt stress effects and the role of salicylic acid in the control of salt stress.

· If possible, add photographs of the plants showing variation in the physiological attributes with respect to varying conductivity and SA concentration.

· Include recently published literature from 2023 on salicylic acid impacts on plants under normal and stressful conditions.

Minor editing is required.

Author Response

Campina Grande, PB

Jul, 18, 2023

Reference: Plants - 2503497 - Response to Review Report 1

Dear Editor

The authors are grateful to you and the unanimous Reviewers for the positive and constructive comments and suggestions on our manuscript entitled “Salicylic Acid as a Salt Stress Mitigator on Chlorophyll Fluorescence, Photosynthetic Pigments, and Growth of Precocious-Dwarf Cashew in the Post-grafting Phase”. The authors would like to inform you that a thorough revision of the manuscript was made, incorporating the suggestions and adopting the text according to the comments. Attached is the revised version of the manuscript. All changes in the text are highlighted in red color.

The authors remain at your disposal for any further information and explanation.

The responses/clarifications to the issues raised by the Reviewer 1/Editor are presented below:

REVIEWER 1

The manuscript reports the positive influence of salicylic acid in attenuating salt stress effects on some physiological attributes. Salicylic acid is now recognized as a potential plant growth regulator for promotion of photosynthesis and metabolites production and resistance to abiotic stress factors. Authors have justified the study by selecting the appropriate concentration of salicylic acid and salt stress. The manuscript, in particular has emphasized the positive role of salicylic acid through proper regression analyses. The presentation of the manuscript can be improved by simplifying the text and expression of results.  Following points may be considered for improvement.

  1. Please give more explanation for the clarity of the experiment, and mention the hierarchy of the Treatments.

Response: Following the reviewer's suggestion, in the revised version of the manuscript, a new table (Table 6) was inserted with a detailed description of the treatments and a sketch (Figure 12) with details of the distribution of experimental units in the experimental area.

  1. Mention in the abstract that the electrical conductivity itself is the salt source (mention if otherwise).

Response:  Dear Reviewer, electrical conductivity is the correct way to express the concentration of salts in a solution.

  1. Clearly state the control setup for each experiment and ensure uniformity in the result.

Response: The control treatment has been described clearly in Table 6 as requested.

  1. The results should be free from ambiguity and try to simplify it, easier to understand to those who are not adept in statistics. Try to emphasize the impact of salt stress effects and the role of salicylic acid in the control of salt stress.

Response:  Dear Reviewer, in the discussion of the results, emphasis was placed on the impact of salt stress and the role of salicylic acid in mitigating such effects, as can be observed in the lines:

  • 109 and 114; 125 and 134; 149 and 158; 161 and 166; 175 and 177; 181 and 184; 196 and 201; 218 and 226; 234 and 238; 249 and 252; 259 and 265; 283 and 287; 297 and 304; 310 and 313; 325 and 331; 357 and 368.

  1. If possible, add photographs of the plants showing variation in the physiological attributes with respect to varying conductivity and SA concentration.

Response: As suggested, photographs (Figures 9 and 10) were inserted in the revised version of the manuscript.

  1. Include recently published literature from 2023 on salicylic acid impacts on plants under normal and stressful conditions.

Response:  As suggested by the Reviewer, articles published in 2023 showing the effects of salicylic acid on plants under salt stress were included in the revised version of the manuscript.

Yours sincerely,

Geovani Soares de Lima

Reviewer 2 Report

Dear Authors!

The increase of plant resistance and productivity under such abiotic stress as salinity by SA treatment is very important issue.

My comments and suggestions:

1) Extensive editing of English required.

2) It is difficult to understand the number of ECw points and SA concentrations in the article. Insert the table listing all treatment options at the beginning of Results and Discussion or in Material and Methods.

3) The text often contains words :"irrigated with electrical conductivity levels" (lines 68, 109, 136, 158, 162, etc.). Rephrase these sentences. It should be clear that plants irrigated with water solutions. The different conductivity is achieved by changing the concentrations of sodium, magnesium and calcium chlorides in water solutions.

4) Change the term saline levels to salinity (Tables) or specify salinity levels more precisely.

5) List SA concentrations ( line 128). It should be mentioned that SA has favorable effect on plant growth and metabolism in small concentrations. Relatively big concentrations inhibit growth and cause physiological disorders. You wrote about it in Conclusion. Add a few words about this in Results and Discussion too. 

6) Rephrase the sentence (lines 151-153). I do not agree that SA reduces chlorophyll content due to the improvement of plant redox metabolism.

7) Rephrase the sentence (lines 347-348). SA can act as a signaling molecule changing the expression of antioxidant genes and influencing the quantity and/or activity of proteins under salinity. 

8) Mention the method and device which help you to evaluate conductivity (page 13).

Recommendations for article design:

1) Change the sentence (line 72) to "... salinity in strawberry plants irrigated by 50 mM NaCl solution".  Change IEL to EL (line 89). Change salicylic acid SL to SA (Table 5).

2) Decipher for the first time ECw (line 91).

3) The abbreviations ROS (line 118), SA ( lines 257, 261, 300, 327), VVI (line 262), SL (line 326), PH (line 513) have already been deciphered.

Author Response

Campina Grande, PB

Jul, 18, 2023

Reference: Plants - 2503497 - Response to Review Report 2

Dear Editor

The authors are grateful to you and the Reviewers for the positive and constructive comments and suggestions on our manuscript entitled “Salicylic Acid as a Salt Stress Mitigator on Chlorophyll Fluorescence, Photosynthetic Pigments, and Growth of Precocious-Dwarf Cashew in the Post-grafting Phase”. The authors would like to inform you that a thorough revision of the manuscript was made, incorporating the suggestions and adopting the text according to the comments. Attached is the revised version of the manuscript. All changes in the text are highlighted in red color.

The authors remain at your disposal for any further information and explanation.

The responses/clarifications to the issues raised by the Reviewer 2/Editor are presented below:

REVIEWER 2

Dear Authors!

The increase of plant resistance and productivity under such abiotic stress as salinity by SA treatment is very important issue.

My comments and suggestions:

  1. Extensive editing of English required.

Response: A thorough revision of the manuscript was made incorporating suggestions from the Editor and unanimous Reviewers.

  1. It is difficult to understand the number of ECw points and SA concentrations in the article. Insert the table listing all treatment options at the beginning of Results and Discussion or in Material and Methods.

Response: Following the Reviewer's suggestion, a new table (Table 6) with the description of the studied treatments was inserted in item 3.2 (Treatments and experimental design), as can be observed in the revised version of the manuscript.

  1. The text often contains words: "irrigated with electrical conductivity levels" (lines 68, 109, 136, 158, 162, etc.). Rephrase these sentences. It should be clear that plants irrigated with water solutions. The different conductivity is achieved by changing the concentrations of sodium, magnesium and calcium chlorides in water solutions.

Response: The excerpts were reformulated according to the reviewer's suggestion..  

  1. Change the term saline levels to salinity (Tables) or specify salinity levels more precisely.

Response: , As suggested, the term was changed in the revised version of the manuscript.

  1. List SA concentrations (line 128). It should be mentioned that SA has favorable effect on plant growth and metabolism in small concentrations. Relatively big concentrations inhibit growth and cause physiological disorders. You wrote about it in Conclusion. Add a few words about this in Results and Discussion too.

Response: Salicylic acid concentrations were inserted in the revised version of the manuscript, as can be seen in line 137. Following the reviewer's suggestions, the text between lines 357 and 361 was modified by mentioning the beneficial effect of salicylic acid depends on the applied concentration.

  1. Rephrase the sentence (lines 151-153). I do not agree that SA reduces chlorophyll content due to the improvement of plant redox metabolism.

Response:  The sentence was reformulated in the revised version of the manuscript (lines 161 and 163).

  1. Rephrase the sentence (lines 347-348). SA can act as a signaling molecule changing the expression of antioxidant genes and influencing the quantity and/or activity of proteins under salinity.

Response: As suggested by the Reviewer, the excerpt was reformulated in the revised version of the manuscript.

  1. Mention the method and device which help you to evaluate conductivity (page 13).

Response:  As suggested by the Reviewer, the information was inserted in the revised version (lines 559-561) of the manuscript.

Recommendations for article design:

  1. Change the sentence (line 72) to "... salinity in strawberry plants irrigated by 50 mM NaCl solution". Change IEL to EL (line 89). Change salicylic acid SL to SA (Table 5).

Response: As per suggestion, the mentioned excerpts were reformulated in the revised version of the manuscript.

  1. Decipher for the first time ECw (line 91).

Response:  The acronym ECw was deciphered as per suggestion in line 92 of the revised version of the manuscript

  1. The abbreviations ROS (line 118), SA (lines 257, 261, 300, 327), VVI (line 262), SL (line 326), PH (line 513) have already been deciphered.

Response: Necessary corrections were made in the revised version of the manuscript, with regard to these excerpts.

Yours sincerely,

Geovani Soares de Lima

Round 2

Reviewer 1 Report

Authors have satisfactorily addressed the concerns made on the earlier version of the manuscript.

Minor editing and spell check required.

Author Response

Campina Grande, PB

Jul, 22, 2023

Reference: Plants - 2503497 - Response to Review Report 1

Dear Editor

The authors are grateful to you and the unanimous Reviewers for the positive and constructive comments and suggestions on our manuscript entitled “Salicylic Acid as a Salt Stress Mitigator on Chlorophyll Fluorescence, Photosynthetic Pigments, and Growth of Precocious-Dwarf Cashew in the Post-grafting Phase”. The authors would like to inform you that a thorough revision of the manuscript was made, incorporating the suggestions and adopting the text according to the comments. Attached is the revised version of the manuscript. All changes in the text are highlighted in red color.

The authors remain at your disposal for any further information and explanation.

The responses/clarifications to the issues raised by the Reviewer 1/Editor are presented below:

REVIEWER 1

Authors have satisfactorily addressed the concerns made on the earlier version of the manuscript.

Response: We are happy to have met the expectations of the reviewer/editor and thank you immensely for the valuable suggestions for improvements.

Yours sincerely,

Geovani Soares de Lima

Reviewer 2 Report

Dear Authors!

1) I did not understand the word samr (line 46).

2) Add "...processes of plants,..." (line 68).

3) I think article a is not needed (lines 117, 141, 166, 201, 210, 226, 289, 313, 332).

4) Add SA "... concentration of SA 1.1 mM " (line 123), "... application of SA 1mM" (line 178)

5) Check the words affect traits (line 137). 

6) Add water '... irrigated with water with highest ..." (line 170).

7) Add VVI (line 315).

8) Check line 360.

Author Response

Campina Grande, PB

Jul, 22, 2023

Reference: Plants - 2503497 - Response to Review Report 2

Dear Editor

The authors are grateful to you and the Reviewers for the positive and constructive comments and suggestions on our manuscript entitled “Salicylic Acid as a Salt Stress Mitigator on Chlorophyll Fluorescence, Photosynthetic Pigments, and Growth of Precocious-Dwarf Cashew in the Post-grafting Phase”. The authors would like to inform you that a thorough revision of the manuscript was made, incorporating the suggestions and adopting the text according to the comments. Attached is the revised version of the manuscript. All changes in the text are highlighted in red color.

The authors remain at your disposal for any further information and explanation.

The responses/clarifications to the issues raised by the Reviewer 2/Editor are presented below:

REVIEWER 2

Dear Authors!

  1. I did not understand the word samr (line 46).

Response: The word has been corrected in the revised version of the manuscript, as can be seen on line 46.

  1. Add "...processes of plants,..." (line 68).

Response: The word was added as suggested, and can be seen on line 68.

  1. I think article a is not needed (lines 117, 141, 166, 201, 210, 226, 289, 313, 332).

Response: Dear reviewer, the tables and figures need to be self-explanatory, the removal of the mentioned excerpts may make it difficult for the reader to understand, for this reason, the excerpts were not removed.

  1. Add SA "... concentration of SA 1.1 mM " (line 123), "... application of SA 1mM" (line 178)

Response: The acronym was added in the mentioned passages, as suggested.

  1. Check the words affect traits (line 137).

Response: The word was corrected in the revised version of the manuscript.

  1. Add water '... irrigated with water with highest ..." (line 170).

Response:  The word was added in the revised version of the manuscript, as can be seen on line 171.

  1. Add VVI (line 315).

Response: The acronym was added in the mentioned passages, as suggested

  1. Check line 360.

Response: The space between paragraphs has been removed in the revised version of the manuscript.

Yours sincerely,

Geovani Soares de Lima
